# The Role of Heavy Metals in the Biology of Female Cancers

**DOI:** 10.3390/ijms26115155

**Published:** 2025-05-28

**Authors:** Joanna Kozak

**Affiliations:** Department of Basic Medical Sciences, Institute of Medical Sciences, Faculty of Medicine, The John Paul II Catholic University of Lublin, ul. Konstantynów 1F, 20-708 Lublin, Poland; joanna.kozak@kul.pl; Tel.: +48-81-454-57-13

**Keywords:** heavy metals, metalloids, breast cancer, ovarian cancer, endometrial cancer

## Abstract

Heavy metals are naturally occurring elements that have numerous applications in industries, agriculture, and other sectors, leading to their widespread distribution in the environment. The constant emission of heavy metals into the environment raises concerns about their impact and harmful effects on living organisms, including human health. Key threats arise from exposure to heavy metals such as lead, cadmium, mercury, and arsenic, all of which are classified as carcinogens. Chronic exposure and bioaccumulation of these metals can result in toxic effects on various body systems, including the female reproductive system. Notably, heavy metals can induce oxidative stress, generate excessive reactive oxygen species, and impair antioxidant defense systems. These metals may also lead to DNA damage, enzyme inactivation, and epigenetic modifications, ultimately disrupting critical cellular processes such as growth, proliferation, differentiation, repair, and apoptosis. Furthermore, some heavy metals can mimic endogenous estrogens, interact with estrogen receptors, and cause hormonal disruptions, a mechanism particularly relevant to the pathogenesis of female-related cancers. Despite significant advances, many gaps remain in our understanding of the molecular mechanisms by which heavy metals contribute to cancer development. Addressing these gaps could facilitate the development of more effective strategies for the prevention and treatment of female cancers. This review highlights the potential effects of heavy metals on molecular pathways in female cancers, suggesting several mechanisms of cancer development.

## 1. Introduction

### 1.1. Heavy Metals

Heavy metals, defined as metallic elements with a relatively high density compared to water, are one of the most extensively studied groups of environmental contaminants. Heavy metals pose significant risks to human health, particularly with increased exposure due to industrial and anthropogenic activities over the last century. Contaminated air, water, and food often lead to acute or chronic poisoning [1,2,3]. Furthermore, heavy metals can accumulate in various body tissues and organs, including the brain, liver, gastrointestinal tract, skin, breast tissue, and gynaecological system [4,5,6,7]. The heavy metals group includes toxic metals such as cadmium (Cd), lead (Pb), manganese (Mn), and chromium (Cr). Additionally, metalloids such as arsenic (As) and light metal aluminium (Al) are also known carcinogens [8]. Arsenite, cadmium, chromium, and nickel (Ni) are classified as Group 1 carcinogens by the International Agency for Research on Cancer (IARC) [9]. Heavy metals may frequently interact with biological systems by losing one or more electrons, forming metal cations with a strong affinity for nucleophilic sites of vital macromolecules. Therefore, they have the potency to disrupt critical cellular processes such as growth, proliferation, differentiation, and apoptosis [10,11]. The speculated carcinogenic mechanisms of heavy metals and metalloids include the induction of oxidative stress and generation of reactive oxygen species (ROS), which damage cellular components, including DNA, proteins, and lipids. Moreover, exposure to heavy metals can initiate epigenetic mechanisms, leading to changes in gene expression without altering the DNA sequence. This can disrupt cellular function and promote carcinogenesis. Chromium, cadmium, and arsenite, in particular, are known to cause genomic instability and impair DNA repair mechanisms, contributing to their carcinogenicity. Additionally, heavy metals can mimic endogenous estrogens, suggesting their role as endocrine disruptors [12,13,14,15]. In this context, heavy metals and metalloids are categorized as metalloestrogens, a subclass of xenoestrogens that, along with other environmental estrogens such as phytoestrogens, possess the ability to bind to estrogen receptors and induce their activation. Moreover, xenoestrogens may influence estrogen signaling not only by receptor binding but also by altering endogenous estrogen levels, antagonizing receptor action, or disrupting estrogen synthesis and metabolism [16,17]. A growing body of evidence shows that heavy metals contribute to the onset and progression of various cancers, including those affecting the female reproductive system and the breast. In the following section, I summarize the current understanding of the contribution of heavy metals and metalloids to the pathogenesis of tumors in the female reproductive system and breast tissue. Despite growing awareness of the dangers posed by heavy metals, the precise mechanisms of their carcinogenicity remain unclear, raising challenging questions about their role in tumor biology.

In summary, heavy metals are well-established environmental contaminants with the potential to disrupt key cellular processes through oxidative stress, DNA damage, epigenetic alterations, and endocrine interference. Some, like cadmium and arsenic, are recognized carcinogens and may act as metalloestrogens. Although their general toxicity is well documented, the exact mechanisms underlying their carcinogenic effects, particularly in the context of hormone-sensitive tissues, require further investigation.

### 1.2. Heavy Metals and Female Cancers

Female cancers, including ovarian, endometrial, and cervical cancers, are associated with substantial morbidity and mortality worldwide, although their prevalence, morbidity, and mortality vary considerably among different cancer types. Epidemiological data from GLOBOCAN 2022 demonstrate that the incidence and mortality patterns of female cancers vary substantially across different regions and cancer types. Ovarian and endometrial cancers show higher incidence rates in industrialized countries, which may be attributed to lifestyle and reproductive factors prevalent in these societies. In contrast, cervical cancer exhibits a markedly higher incidence and mortality rate in low- and middle-income countries, primarily due to inadequate access to effective screening programs and HPV vaccination initiatives. Breast cancer is the second most common cancer in woman. Although the results of breast cancer treatment are improving, the disease negatively affects the quality of life of hundreds of thousands of women around the world. A growing body of evidence suggests that heavy metals and metalloids are associated with the risk of developing various cancers, including female cancers.

### 1.3. Breast Cancer

Among the plethora of speculated risk factors for breast cancer, several key contributors include early age at menarche, late age at menopause, older age at first full-term pregnancy, hormone replacement therapy (HTR) for menopausal symptoms, and obesity [15]. Moreover, a positive family history of breast or ovarian cancer remains one of the strongest predictors of increased risk. Additionally, pathogenic variants in genes such as BRCA1, BRCA2, TP53, PALB2, and CHEK2 significantly elevate the lifetime risk of breast cancer [18,19,20]. Beyond genetic predisposition, a personal history of benign breast diseases—particularly atypical ductal hyperplasia, atypical lobular hyperplasia, and lobular carcinoma in situ—has also been recognized as a notable risk factor [21]. Another suggested risk factor is exposure to metals and metalloids with estrogen-like activity [15]. It is thought that the most significant risk factors for developing breast cancer are associated with increased lifetime exposure to estrogens, whether endogenous or exogenous [22]. Exogenous estrogens include metalloestrogens, which are metals that can activate estrogen receptor-alpha (ERα) in the absence of estradiol. These metals and metalloids fall into two categories: bivalent cations such as cadmium, calcium (Ca), lead, mercury (Hg), cobalt (Co), copper (Cu), nickel, chromium, and tin; oxyanions such as arsenite, nitrite, selenite, and vanadate [15,22]. ERα is a ligand-inducible transcription factor belonging to the nuclear receptor superfamily. It plays a key role in regulating gene transcription upon hormone binding [16]. The cloning of different members of this family has led to the identification of several functionally related domains. Region C, which contains two zinc fingers, is involved in DNA binding, with the first zinc finger determining DNA-binding specificity and the second stabilizing the interaction with DNA [16]. Regions A/B, located at the N-terminal end, possess transactivation function-1 (AF-1), which plays a role in both ligand-dependent and independent receptor activation. Region D, known as the hinge region, provides flexibility to the receptor structure, while Region E, the ligand-binding domain (LBD), is responsible for ligand-dependent activation of AF-2. This region is also crucial for receptor dimerization and interaction with coactivators and corepressors. Finally, the function of Region F remains less well defined compared to other regions [23,24,25]. The structure and functions of the major domains of ERα are illustrated and summarized in Figure 1.

Several studies have highlighted the effect of heavy metals on the structure and function of steroid receptors. Metals such as cadmium, selenite, cobalt, copper, nickel, chromium, lead, mercury, tin, and vanadate can interact with the hormone-binding domain and block the binding of estradiol to the receptor [26]. In the late 1990s, the discovery of estrogen receptor beta (ERβ) introduced a new dimension to estrogen signaling [27]. ERβ is highly homologous to ERα but differs in tissue distribution and functional roles. While the DNA-binding domain of ERβ is highly conserved with that of ERα, sharing 96% similarity, the ligand-binding domain is only 58% conserved, suggesting potential differences in ligand specificity and receptor activation mechanisms. Additionally, the less conserved A/B and F domains indicate potential differences in receptor modulation and interactions with coactivators or corepressors [27]. ERα is predominantly found in the uterus, mammary gland, testis, pituitary, liver, kidney, heart, and skeletal muscle, whereas ERβ is significantly expressed in organs such as the ovary and prostate, and in tissues such as the endometrium [28]. This differential expression pattern suggests that ERβ may play distinct roles in these tissues, potentially influencing estrogen-dependent physiological and pathological processes differently from ERα. ERα is predominantly associated with oncogenic activity across various cancers. In breast cancer, it is upregulated in approximately 75% of tumors and promotes proliferation via estrogen-mediated signaling [29,30]. ERα overexpression correlates with luminal A subtype and is a major target of endocrine therapies such as tamoxifen, aromatase inhibitors, and fulvestrant [30,31]. Similar oncogenic effects are observed in prostate, ovarian, lung, colon, and liver cancers, where ERα activation enhances tumor growth, migration, and the epithelial-mesenchymal transition (EMT) process, and resistance to apoptosis [32,33].

ERβ, by contrast, is commonly described as a tumor suppressor [34,35]. Its expression is typically lower in malignant tissues and is inversely correlated with ERα levels [36]. In ERα-positive breast cancer, ERβ downregulates ERα activity, thereby inhibiting tumorigenesis [37]. In prostate cancer, ERβ exerts anti-tumor effects by upregulating pro-apoptotic genes such as p53 upregulated modulator of apoptosis (PUMA), and by inhibiting the hypoxia-inducible factor 1-alpha (HIF-1α) pathway [38,39]. Its suppression of EMT via upregulation of E-cadherin further reinforces its role as a tumor suppressor [40]. In other malignancies, such as ovarian and colorectal cancer, ERβ similarly acts to inhibit tumor progression through the regulation of cell cycle checkpoints and apoptotic mechanisms [41,42]. In hepatocellular carcinoma (HCC), ERβ attenuates pro-tumor inflammatory signaling by inhibiting the JAK/STAT6 pathway and tumor-associated macrophages polarization [43]. However, ERβ activity is not universally suppressive. In certain contexts, such as ERα-negative breast tumors and advanced prostate cancer, ERβ isoforms can form heterodimers (e.g., ERβ1 with ERβ2 or ERβ5) that contribute to increased malignancy [44,45]. These nuanced, isoform-specific functions underscore the importance of context in evaluating ERβ as a therapeutic target.

In summary, ERα and ERβ represent functionally divergent nuclear hormone receptors with critical roles in the development and progression of sex hormone-dependent cancers. ERα predominantly drives oncogenic pathways, whereas ERβ exerts tumor-suppressive effects, although exceptions exist depending on isoform composition and cellular context. Understanding the dynamic interplay between these receptors is essential for refining hormone-based therapeutic approaches and for developing precision medicine strategies tailored to individual receptor expression profiles. However, a detailed discussion of this topic is beyond the scope of this article.

Environmental exposure to arsenite and cadmium has demonstrated estrogen-like activity in human breast cancer cell lines (MCF-7). Furthermore, in vivo studies using Sprague–Dawley rats have confirmed that environmentally relevant doses of arsenite and cadmium promote mammary tumor development. In experiments conducted on ER-positive MCF-7 breast cancer cells, arsenite was found to interact with the hormone-binding domain of ERα, blocking estradiol binding and significantly stimulating the growth of MCF-7 cells [15,46]. The proliferation of these arsenite-stimulated cells may be attributed to intracellular effects on signaling pathways such as the mitogen-activated protein kinase (MAPK), phosphatidylinositol 3-kinase (PI3K), Raf-MEK-ERK1/2, and protein kinase B (Akt) pathways [47]. Further studies on MCF-7 breast cancer cells and ovariectomized animals showed that arsenite and cadmium increased the global expression of estrogen-responsive genes such as the progesterone receptor (PgR), growth regulation by estrogen in breast cancer 1 (GREB1), and c-fos in the mammary gland, as well as the expression of complement C3, c-fos, and cyclin D1 in the uterus [15]. Moreover, experiments using a DMBA-induced rat mammary tumor model showed that a diet supplemented with arsenite and cadmium significantly increased the incidence of mammary tumors and decreased the latency to tumor onset, without affecting tumor multiplicity or total tumor volume [15]. Taken together, these experimental data show that environmentally relevant amounts of arsenite and cadmium promote mammary tumorigenesis, which may be partially caused by their ability to activate ERα and, consequently, activate downstream signaling pathways that stimulate cell proliferation and influence other cellular functions. Parodi and colleagues conducted detailed investigations to evaluate the effects of prenatal arsenic exposure on mammary gland development, with particular emphasis on the expression and regulation of ERα [22]. Their in vivo experiments demonstrated that in female offspring, in utero exposure to an environmentally relevant dose of arsenite significantly advanced the timing of vaginal opening and altered mammary gland development prior to affecting the hypothalamic-pituitary-gonadal axis [22]. Furthermore, arsenite exposure was associated with an expansion of the mammosphere-forming cell population in the prepubertal mammary gland and induced the overexpression of ERα transcripts, accompanied by altered regulation of their expression in response to estradiol, in the postpubertal gland. These effects of arsenite exposure could potentially increase the gland’s susceptibility to neoplasia later in life [22]. These observations, along with epidemiological studies on the influence of phytoestrogens and the xenoestrogen polybrominated biphenyl on early puberty onset, suggest that exposure to environmental estrogens may be linked to breast cancer development [48,49,50].

Beyond estrogen-like activity, metals and metalloids can also modulate cellular signaling pathways independently of estrogen receptor activation. Oxidative stress, characterized by an overproduction of ROS that overwhelms the cellular antioxidative system, can severely damage membrane lipids, proteins, and DNA. This relationship between heavy metals, ROS, and breast cancer has been explored in various studies [51,52]. For instance, Cannino and colleagues reported that cadmium exerted cytotoxic effects on MDA-MB231 ER-negative breast cancer cells, suggesting a mechanism different from ER activation. Their results showed that only 96-h exposure to cadmium chloride (CdCl_2_) induced a massive accumulation of ROS, even without changes in mitochondrial transmembrane potential [53]. Moreover, the study revealed that cadmium chloride exposure actively modified the expression levels of heat shock protein (hsp60) and cytochrome oxidase (COX) subunits II and IV. COX is the terminal component of the mitochondrial respiratory chain, with subunits II encoded by the mitochondrial genome and IV encoded by a nuclear gene. The induction of hsp60 is expected to assist the cell in refolding and processing damaged proteins or regulating programmed cell death [53]. Using the same cell line, it was shown that cadmium has the potency to activate the p38/MAPK signalling pathway. p38/MAPK is involved in normal cell physiology, controlling chromatin remodelling, DNA methylation and transcriptional patterns, as well as regulating cell cycle, its checkpoints, and decisions related to cell life or death [54,55]. Another interesting activity of cadmium was demonstrated by Liu et al. Their results showed that in MCF-7 cells, a relatively benign breast cancer cell line, cadmium rapidly activated extracellular signal-regulated kinase 1 and 2 (ERK1/2), AKT, and ERα at a micromolar concentration within 2.5 min. Under the same exposure condition, SK-BR-3 cells, a moderately malignant breast cancer cell line, responded more slowly, with ERK1/2 activation occurring at 7.5 min. In comparison, the highly invasive breast cancer cell line MDA-MB-231 did not exhibit a rapid response to cadmium treatment at all. The dramatic differences in response between these three cell lines suggest that cadmium sensitivity may be associated with the relative invasiveness of the breast cancer cells [56]. Both ERK1/2 and AKT signalling regulate fundamental cellular processes in response to external stimuli, including certain chemicals or physical agents, one of which could be cadmium. These processes include cell proliferation, survival, growth, metabolism, migration, and differentiation [57]. Cadmium-induced disruption of the ERK1/2 and AKT signalling pathways may lead to the transformation of cells toward a tumour phenotype. Continued cadmium exposure in cells with a fully developed tumor phenotype does not cause changes in ERK1/2 and AKT activity for unknown reasons.

In line with previous reports on the potential of heavy metals to induce neoplastic transformation, two independent experiments have demonstrated the intriguing effects of arsenite. In certain models, arsenite has been shown to restore ERα expression in ER-negative breast cancer cells [58]. However, in the study using the ER-negative normal breast epithelial cell line MCF-10A, chronic exposure to a low dose of arsenite (500 nM) for 24 weeks resulted in the acquisition of a cancer cell phenotype through an ERα-independent mechanism [59]. These arsenic-exposed breast epithelial (CABE) cells exhibited increased metalloproteinase activity, enhanced colony formation, invasion, and proliferation rates [58]. The CABE cells remained HER-2 and PgR negative and displayed strong expression of cytokeratin 5 (K5) and p63. Although not meeting the full criteria for basal-like or triple-negative breast cancer due to the detectable, albeit low, expression of ERα, their molecular profile shows several features typically associated with these aggressive breast cancer subtypes [59,60]. Furthermore, CABE cells exhibited a loss of contact inhibition and formed multilayers, suggesting that an epithelial-to-mesenchymal transition (EMT) occurred during chronic arsenic exposure [59]. In the second experiment, Du and co-workers demonstrated that arsenic trioxide induces re-expression of ERα in ER-negative breast cancer cells through demethylation of the ERα promoter, indicating that arsenic trioxide acts through an epigenetic mechanism. These findings highlight the potential use of arsenic trioxide in anti-cancer therapy for treating ER-negative breast cancer. The restoration of ERα expression by arsenic trioxide is sufficient to induce anti-estrogen responses in both ER-negative breast cancer cells and animal models [61].

Contrary to these findings, epidemiological studies investigating the potential link between cadmium, a toxic metal found in the environment and certain foods, and the risk of breast, endometrial, or ovarian cancers in postmenopausal women have shown no clear association. These three cancers are the most frequent hormone-related cancers among women and are often analysed together [62]. Most of these studies involved postmenopausal women who provided detailed dietary information through food frequency questionnaires, with cadmium intake estimated from these reports. In two independent studies, Adams and colleagues did not observe that dietary cadmium is a risk factor for breast cancer in postmenopausal women [63,64]. Similarly, Eriksen et al. found no significant associations between dietary cadmium intake and risk of hormone-related cancers in postmenopausal women [65]. In line with these results, two Japanese studies also failed to find an association between dietary cadmium intake and breast or endometrial cancer [66,67]. Given the aforementioned studies, it remains unclear how to explain the discrepancy between laboratory evidence suggesting carcinogenic potency of cadmium and the lack of significant associations between dietary cadmium intake and risk of hormone-related cancers in postmenopausal women in epidemiological studies. We speculate that the differences may stem from the unique methodologies used in each approach. In laboratory experiments, all conditions are fully controlled, and results are directly connected to the experimental variables. In contrast, epidemiological studies rely on questionnaires, where researchers have limited control over experimental conditions. Additionally, in vitro studies often use relatively high concentrations of the metal over short exposure times, which can be considered “acute” compared to real-life human exposure. In reality, metal exposure tends to be low but persistent, with metals accumulating in different tissues over the years, potentially causing long-term negative effects. Overall, both types of studies are highly valuable and important, offering opportunities to explore this research area more thoroughly.

In summary, metal exposure occurs primarily through environmental contamination of food, groundwater, drinking water, air, and soil. A wealth of data indicates that heavy metals and metalloids can activate ERα, mimicking the actions of physiological estrogens and stimulating ER effectors in breast cancer cell lines. Significant evidence suggests that cadmium, arsenite, and lead modulate intracellular signalling pathways, altering cellular phenotypes toward cancerous behavior. Therefore, it seems reasonable to put an effort into further research to elucidate the role of metals and metalloids in breast cancer biology.

### 1.4. Ovarian Cancer

Although ovarian cancer is less frequently diagnosed compared to other gynaecological malignancies, it is characterized by a disproportionately high mortality rate [62]. The poor prognosis associated with ovarian cancer is primarily due to its late diagnosis at advanced stages, where treatment options are very limited. Another reason for the unfavourable prognosis is the nonspecific nature of medical symptoms, which often go unrecognized in the early stages of the disease [68]. In terms of etiology, some risk factors are linked to nutritional habits, such as obesity, low consumption of fruits and vegetables (and therefore low fiber intake), as well as occupational and environmental exposure to xenoestrogens [68,69]. Additional well-established risk factors include age, family history of ovarian cancer, and infertility treatment (particularly hormonal stimulation), while oral contraceptive use has been shown to reduce the risk of ovarian cancer [70]. Epidemiological studies have highlighted an increased risk of ovarian cancer associated with estrogen and progesterone therapy, especially estrogen-only therapy [71]. Because ovarian cancer is predominantly a disease of industrialized countries, xenoestrogens have been suspected of contributing to the rising incidence of this cancer. Therefore, understanding the role of these compounds in ovarian cancer development has important implications for disease prevention. Xenoestrogens, including metalloestrogens, have the potential to disrupt hormonal homeostasis, increasing the risk of hormone-dependent cancers like ovarian cancer. Although data on the role of metalloestrogens in ovarian cancer are limited, the potential links between these compounds and hormones warrant further investigation.

An epidemiological study by García-Pérez et al. found a statistically significant excess of ovarian cancer mortality among women living near Spanish industrial sites [70]. The elevated risks were concentrated around facilities that release metals classified as Group 1 carcinogens by the IARC, including arsenic, cadmium, chromium, and nickel compounds [70]. Cadmium, in particular, has been the subject of many studies exploring its carcinogenic potential in ovarian cancer, though epidemiological research presents conflicting results [72,73]. Despite the inconsistencies, the importance of cadmium as a potential risk factor for hormone-related cancers remains evident. The role of other heavy metals in ovarian cancer has also been examined. Canaz et al. compared tissue concentrations of lead, nickel and selenium in healthy non-neoplastic ovaries (*n* = 20), epithelial ovarian cancer (*n* = 20), and epithelial borderline tumors (*n* = 15) that carry an intermediate behaviour between benign and frankly malignant neoplasms both in clinical and histopathological aspects [74]. Their findings revealed higher concentrations of lead and nickel in both malignant and borderline ovarian tumors compared to healthy ovaries. The limitations of this study are its retrospective design, analysing only three elements in a limited number of patients, and a comparison of the elements in paraffin-embedded tissue samples, not in fresh tissues [74]. Although the study had limitations, the results suggest that the accumulation of these metals in ovarian tissue may be associated with increased proliferation rates in borderline or malignant epithelium In another study, Sawicka and colleagues assessed the interaction between cadmium, 17β-estradiol (E2), and its metabolites (2-methoxyestradiol [2-MeOE2] and 16α-hydroxyestrone [16α-OHE1]) on the viability and P-glycoprotein (P-gp) expression levels in the cisplatin-resistant SKOV-3 ovarian cancer cell line [69]. The study included two experimental models: simultaneous exposure of cells to cadmium and estrogens, and pre-incubation with estrogens prior to cadmium exposure. Cell viability was assessed using the MTT assay, and the type of interaction was determined by calculating the Combination Index (CI) with CompuSyn software. P-gp expression was evaluated immunocytochemically. E2 is known to promote ovarian cancer cell proliferation by inhibiting cell-to-cell adhesion, which may contribute to increasing cancer risk and metastasis, while 2-MeOE2 is considered to have antitumor activity, and 16α-OHE1 is thought to exert carcinogenic effects [69,75,76]. P-gp, a transmembrane protein belonging to the ATP-binding cassette (ABC) transporter subfamily B, is commonly upregulated in multidrug-resistant ovarian neoplasm [77]. Sawicka and colleagues demonstrated that exposure of cisplatin-resistant ovarian cancer cells to cadmium resulted in increased expression of P-glycoprotein (P-gp), a key efflux transporter associated with multidrug resistance. These findings suggest that environmental cadmium exposure may exacerbate chemotherapy resistance in ovarian cancer, potentially leading to poorer treatment outcomes [69]. Complementary research by Wieder-Huszla et al. investigated the serum concentrations of elements such as Na, Mg, Ca, Zn, P, Cu, Fe, Cd, Ni, and Sr in 50 patients with advanced endometrial (*n* = 24) and ovarian cancer (*n* = 26) [78]. The authors demonstrated that cadmium concentrations were influenced by the interaction between the type of tumor (ovarian or endometrial cancer) and the stage of chemotherapy, although these findings should be interpreted cautiously due to the limited number of cadmium measurements [78]. Taken together, these studies suggest that cadmium may be involved in both hormonal disturbances and chemotherapy resistance. Acquired chemoresistance is a multifactorial phenomenon influenced by tumor type, stage, and cellular ROS levels [51]. Given that heavy metals also induce oxidative stress and overproduction of ROS, understanding the interplay between ROS and cadmium is crucial to improving cancer treatment outcomes.

In the context of the etiology of ovarian cancer, which in many cases originates from the epithelium of the fallopian tube [79], there appears to be a rationale for examining the role of heavy metals in fallopian tube carcinogenesis. However, to the best of my knowledge, no studies to date have demonstrated a direct association between heavy metal exposure and fallopian tube cancer. This highlights a notable gap in the literature regarding potential interactions between heavy metals and the pathogenesis of this malignancy.

In summary, although there is considerable interest in the role of heavy metals in carcinogenesis, specific data on the molecular mechanisms through which they affect cellular pathways in ovarian cancer remain limited. The accumulation of heavy metals in ovarian tissue, their potential to disrupt hormonal balance, and their possible role in chemotherapy resistance highlight key areas for future investigation. Similarly, the lack of evidence supporting a definitive role of heavy metals in fallopian tube cancer represents a challenging yet essential direction for future research.

### 1.5. Endometrial Cancer

Endometrial cancer (EC) is one of the most common malignancies affecting the female reproductive system [62]. When diagnosed early, EC typically has a favorable prognosis and a high survival rate. However, late-stage EC is associated with poor outcomes, largely due to its aggressive progression, characterized by deep myometrial invasion and lymph node metastasis [80,81,82]. EC is divided into two primary pathogenic types. Type I, which is estrogen-dependent, is frequently associated with obesity and typically has a favourable survival rate. In contrast, Type II is less dependent on estrogen and is relatively rare, constituting only 10% of all EC cases. However, Type II is more aggressive and has a poorer prognosis, accounting for 40% of deaths among EC patients compared to Type I [81,83]. Several risk factors are associated with EC, including a family history of cancer, exogenous estrogen exposure (such as phytoestrogens and hormone-based drugs), obesity, diabetes mellitus, age, and exposure to heavy metals like cadmium [81,84,85,86,87,88,89].

Human endometrial tissue has the potential to accumulate heavy metals such as cadmium, lead, chromium, and nickel, and women who smoke tend to have higher levels of these metals in their bodies [6,90]. As suggested by Rzymski and co-workers, heavy metals, including cadmium, lead, chromium, and nickel, are considered potential metalloestrogens and may contribute to reproductive system disorders like endometriosis and EC [6]. Since it is well known that human endometrium expresses high levels of ERα and ERβ, it is a potential target for interaction with metalloestrogens, specifically cadmium, lead, chromium, and nickel [6,28]. Stoica et al. showed that cadmium activates ERα through interaction with the hormone-binding domain of the receptor in COS-1 cells [46]. Moreover, activated ERα stimulates the transcription of the PgR and pS2 genes, which play essential roles in cell proliferation and differentiation [16]. Based on this, we can speculate that ERα, stimulated by cadmium accumulation in the endometrium, could trigger uncontrolled cell proliferation and contribute to neoplastic transformation.

Another interesting study on the carcinogenic potential of cadmium and lead was conducted by Michalczyk and colleagues. Significant differences in the cadmium-to-lead ratio were observed among 110 patients diagnosed with various pathologies, such as uterine myomas, endometrial polyps, and EC, with the highest cadmium level found in patients with EC [91]. Additionally, their results showed that an increased blood cadmium concentration above the median level is a risk factor for EC, while differences in lead concentration were not significant in the same patient group. As the authors stressed, their results need to be validated in a larger population sample. Nevertheless, the data support the hypothesis that cadmium is an important risk factor in EC pathology.

Further evidence supporting the importance of metalloestrogens in EC biology was provided by Guyot and colleagues. They measured the concentration of lead, mercury, cadmium, and vanadium (V) in human endometrial tissue samples from individuals with hyperplasia or adenocarcinoma, as well as in normal tissues. They found that hyperplasic endometrial tissue has a four-fold higher concentration of mercury compared to normal tissue. Moreover, in vitro studies on the Ishikawa and HEC-1B EC cell lines showed that mercury, lead, and cadmium may affect the aryl hydrocarbon receptor (AhR) signaling pathway, which regulates the expression of several xenobiotic-metabolizing enzymes, CYP1A1 and CYP1B1 [92]. Guyot and colleagues’ research suggested that mercury dramatically induced morphological changes (cell spreading) and alterations in the cytoskeleton (actin, stress fibres, paxillin) in Ishikawa and HEC-1-B cell lines, although these changes were not part of the EMT program [92]. Furthermore, they showed that mercury did not significantly regulate CYP1A1 and CYP1B1, AhR target genes, in the Ishikawa cell line, unlike cadmium, which had a detectable and statistically significant effect on CYP1A1 expression at the chosen time points [92]. Special interest has been given to the induction of oxidative stress by mercury, cadmium, and lead in human EC cell lines, as manifested by increased expression levels of oxidative stress markers, such as heme oxygenase 1 (HO1) and NAD(P)H quinone oxidoreductase 1 (NQO1). These studies are among the few that focus on the effect of heavy metals on the oxidative stress response in EC cells. According to the research of Wang et al. and Son et al., ROS are considered a key mechanism in cadmium-induced carcinogenesis. They suggest that ROS play a major role in the transformation of human bronchial epithelial cells BEAS-2BR exposed to cadmium during the first stage of cadmium-induced carcinogenesis, transitioning cells from normal to transformed. At the molecular level, in cadmium-transformed cells, p62 and nuclear factor erythroid 2-related factor 2 (Nrf2) were constitutively activated, and their downstream antioxidants and anti-apoptotic proteins were elevated. The final outcomes included a decrease in ROS levels, apoptosis resistance, and tumorigenesis [93]. Moreover, Son et al.’s findings indicate a direct involvement of ROS in cadmium-induced carcinogenesis and implicate the AKT/GSK-3β/β-catenin signaling pathway in this process [94]. These studies suggest that cancer cells activate mechanisms that protect them against excessive ROS, allowing them to escape oxidative stress. This probably leads to the formation of a new oxidative-reductive balance in cancer cells at very early stages of development. The described mechanism is characteristic of human bronchial epithelial cells, but the question remains whether the same mechanism is valid for EC cells.

In summary, research highlights the role of cadmium, lead, and mercury in EC biology. Some studies support the hypothesis that heavy metals can disrupt the estrogen molecular pathways in EC, while others shed light on the role of heavy metals in the oxidative stress signalling pathway. Several studies have shown clearly elevated levels of heavy metals in the endometrium tissue of women with EC. These findings support the hypothesis that heavy metals, particularly cadmium and lead, induce estrogenic effects, which may increase the risk of hormone-related malignancies like EC. Overall, this research underscores the need for further studies on the effects of heavy metals on EC, with a particular focus on understanding the molecular mechanisms involved in oxidative stress and tumor transformation.

### 1.6. Cervical Cancer

Cervical cancer is one of the leading causes of cancer-related death among women worldwide [62]. It is also a cancer that, theoretically, could be eliminated with comprehensive preventive and control strategies [95]. According to epidemiological data from the Global Cancer Observatory (GLOBOCAN 2022), the primary cause of cervical cancer is persistent infection with high-risk human papillomavirus (HPV) types, predominantly HPV16 and HPV18, which are responsible for approximately 70–80% of cases. Behavioral factors such as early sexual initiation, multiple sexual partners, and unprotected sex increase the risk of acquiring an HPV infection. Other cofactors that promote the progression to cancer include smoking, immunosuppression (e.g., HIV infection), unhealthy dietary habits, and environmental exposures such as heavy metals (e.g., lead and cadmium) [62,90]. Zhang et al. showed a correlation between increased cervical tissue lead concentration and the progression of cervical intraepithelial neoplasia (CIN). The study concluded that higher lead levels in cervical tissues are associated with a greater risk for CIN in comparison to controls and non-HPV-infected individuals [96]. Similarly, a study by Rzymski and colleagues found elevated lead and chromium levels in CIN subjects compared to histologically normal tissues. Their research also indicated that both current and former smokers had significantly higher concentrations of cadmium and lead in cervical tissues [5].

In summary, although the current literature directly linking heavy metals to cervical cancer is limited, available studies suggest a possible association, particularly with lead and cadmium accumulation in cervical tissue. Findings indicating higher concentrations of these metals in patients with cervical intraepithelial neoplasia, especially among smokers, support the need for further investigation. Future research should focus on clarifying the potential role and synergistic effects of carcinogenic heavy metals in cervical cancer development, as this remains an underexplored but potentially significant area.

### 1.7. Other Female Cancers

It is reasonable to investigate whether exposure to heavy metals may be linked to the development of vulvar and vaginal cancers, as well as gestational trophoblastic disease (GTD), including choriocarcinoma. To date, there is a lack of data in the literature directly associating specific heavy metals, such as cadmium, lead, mercury, or arsenite, with these malignancies. This remains a largely underexplored area of research. Given that heavy metals are known to cause DNA damage, hormonal disruption, and oxidative stress—the most frequently described mechanisms of their action—it is plausible that they may also contribute to the pathogenesis of vulvar and vaginal cancers or choriocarcinoma via similar molecular pathways. Only a few studies have addressed potential correlations between heavy metal exposure and vulvar or vaginal cancers. Motlhale et al. reported a possible association between the use of smokeless tobacco (SLT), specifically snuff, and an increased risk of these cancers. This risk may be attributed to carcinogenic heavy metals present in SLT products, such as cadmium, nickel, and chromium. The authors proposed that local exposure of genital tissues, particularly through intravaginal use, could exert carcinogenic effects. However, due to the inconclusive nature of the results, the authors recommend cautious interpretation and emphasize the need for further studies focused on usage patterns, exposure duration, and chemical composition of SLT products [97]. A separate case report by Yang et al. described vulvar squamous cell carcinoma (VSCC) in a woman with long-term occupational exposure to heavy metals during copper refining and smelting. The study suggested that such exposure may increase the generation of ROS, leading to glutathione depletion, DNA damage, lipid peroxidation, protein alterations, and cellular disturbances that could promote carcinogenesis. Heavy metals such as arsenite, lead, and cadmium may also exert toxic effects by binding to protein sulfhydryl groups and depleting glutathione levels, thereby impairing antioxidant defenses and compromising membrane integrity. Enzymes such as glutathione peroxidase (GPX) and catalase (CAT), as well as total thiol molecules (TTM), play crucial roles in counteracting ROS. Reduced levels of GPX, CAT, and TTM observed in copper smelter workers suggest decreased antioxidant capacity. There is increasing evidence that oxidative stress contributes to the pathogenesis of squamous cell carcinoma, although the oxidative profile of affected individuals remains incompletely understood [98].

Although no direct evidence currently links heavy metal exposure to the etiology of choriocarcinoma, some studies suggest that cadmium may affect the physiology of human placental choriocarcinoma cell lines, such as BeWo and JEG-3, with potential oncogenic implications. Kummu et al. provided molecular evidence that cadmium may interfere with the function of the ABCG2 transporter, an efflux protein located on the maternal-facing membrane of the placenta, which protects the fetus by transporting xenobiotics from the syncytiotrophoblast to maternal circulation [99]. Other studies using JEG-3 cells also explored cadmium’s effects. While these findings do not confirm a direct causative role, cadmium may act as a carcinogen indirectly through epigenetic modifications that alter the expression of genes involved in proliferation, differentiation, and apoptosis [100].

In summary, the direct impact of heavy metals on vulvar and vaginal cancers and choriocarcinoma, has not been conclusively demonstrated. Nevertheless, indirect mechanisms such as oxidative stress and epigenetic modifications may play a role in their pathogenesis. Preliminary evidence from case studies involving SLT use and occupational exposure in copper smelting highlights potential risk factors. In the context of choriocarcinoma, in vitro studies indicate that cadmium may impair placental defense mechanisms. These findings underscore a significant knowledge gap and support the need for further research on the role of heavy metals in the development of these cancers.

### 1.8. Conclusions

Female cancers are associated with substantial morbidity and mortality worldwide. Heavy metals can accumulate in tissues and organs, progressively altering their biology and promoting carcinogenesis. The primary mechanism by which heavy metals influence the development of female cancers seems to involve their interaction with estrogen receptors and the disruption of related signaling pathways. Additionally, heavy metals can induce oxidative stress and epigenetic alterations, both of which are critical in the process of tumorigenesis. An overview of the complex interaction network between heavy metals and their numerous molecular targets, as summarized in Figure 2, illustrates the multifactorial nature of this influence.

Despite significant advances, many gaps remain in our understanding of the molecular mechanisms through which heavy metals contribute to cancer development. Addressing these gaps could facilitate the development of more effective strategies for the prevention and treatment of female cancers. Given the considerable health risks associated with heavy metals, further research into their role in the pathogenesis of female cancers is essential.

## Figures and Tables

**Figure 1 ijms-26-05155-f001:**
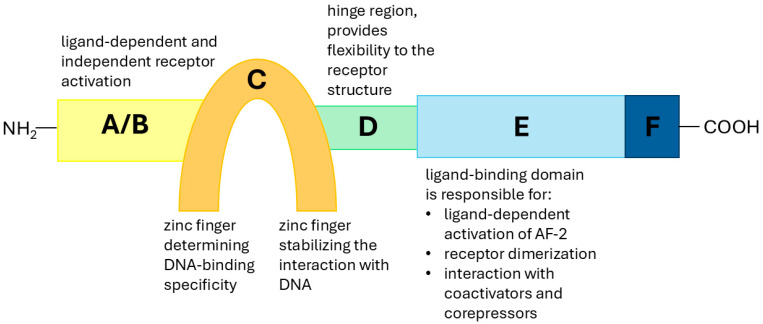
Schematic overview of the structure and major functional domains of ERα. The yellow area corresponds to the A/B domain, responsible for ligand-dependent and independent activation (AF-1). The orange area corresponds to the C domain (DNA-binding domain), containing zinc fingers for DNA interaction. The green area corresponds to the D domain (hinge region), which provides flexibility and nuclear localization signals. The light blue area corresponds to the E domain (ligand-binding domain), responsible for ligand-dependent activation (AF-2), receptor dimerization, and coactivator/corepressor interaction. The dark blue area corresponds to the F domain, which remains less defined relative to other regions.

**Figure 2 ijms-26-05155-f002:**
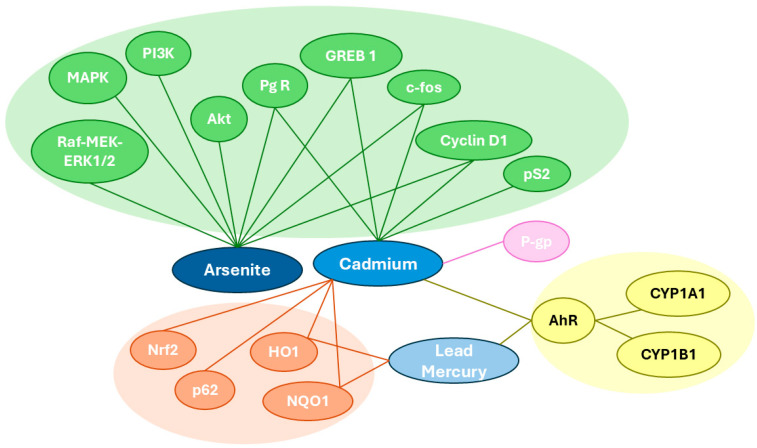
Schematic overview of the interaction network of selected heavy metals. The green area corresponds to cellular pathways mediated by estrogen receptors (ERs). The orange area corresponds to the cellular antioxidant system. The yellow area corresponds to signaling pathways that regulate the expression of xenobiotic-metabolizing enzymes. The violet area represents interactions with transmembrane transport proteins. The dark blue area represents arsenite, the medium blue represents cadmium, and the light blue represents lead and mercury.

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
