# Peer review of "The Role of Heavy Metals in the Biology of Female Cancers"

_ijms, 2025, doi:10.3390/ijms26115155_

Round 1

Reviewer 1 Report

Comments and Suggestions for Authors

The paper is a review on the role of metals in developping gynaecological tumor. 

Various aspects have seen considered and well described.
The article is a review on the role that metals could have in the onset of gynecological tumors.
The work is original also because consider different kind of tumors and well describe the molecular and cellular toxic activity of metals. 
In addition, these kind of tumors, in my opinion, are less reported than others, thus, this paper makes up for this lack.
The fact that it reports information on various kind of gynecological neoplasm and on the metals involved in their development
Conclusions are well describe highlighting the main points of the entire work.

The paper can be accepted in this form after the revision of the reference no. 9 in the list. Name are missing and also other information regarding year, volume, pages, etc., of the reference.

Reviewer 2 Report

Comments and Suggestions for Authors

The study entitled "The role of heavy metals in the biology of gynaecological neoplasm" presented for review consists of 14 pages with 67 references. The manuscript fits the Journal scope. 

The title: Due to the description of the effects of heavy metals on several gynecological cancers I would suggest to change "neoplasms".

Keywords are adequate and refer to the whole context of the article. 

The author focused on 4 basic heavy metals (lead, cadmium, mercury, arsenic) and described potential molecular mechanisms leading to carcinogenesis. 

Line 55: phytoestrogens are also xenoestrogens. 

Line 55-56: They not only bind to receptor. Xenoestrogens may act in various ways. They are capable of re-emerging endogenous estrogens, antagonizing their action, disrupting the synthesis and its metabolism.

Line 58: "In the following section, we summarize" - There is one author or more?

Line 407-411: I would place it directly under the figure as a legend.

Figure 1 is clearly presented. 

Cervical cancer:

-Line 373: Two oncogenic and most common types of the virus: HPV16 and HPV18 are responsible for the development of up to 70-80% of cases. HPV is not a secondary risk factor.

Comments on the Quality of English Language

Line 372: These infection (infections).

Reviewer 3 Report

Comments and Suggestions for Authors

General comments:

The review by Joanna Kozak is potentially interesting but in my opinion it needs improvement. It is a bit chaotic and lacks clear conclusions. First of all, breast cancer is not considered a gynecological cancer. Gynecologic cancer is any cancer that starts in a woman's reproductive organs, including the uterus, ovaries, cervix, vagina and vulva. Thus, the title should be changed (e.g.: The role of heavy metals in the biology of female cancers). Possibly, there should be a short chapter concerning other gynecological cancers (e.g.: Other cancers), concerning vulvar and vaginal cancer, gestational trophoblastic disease (GTD) including choriocarcinoma. Even, if there is little or no data about the role of heavy metals in these cancers, this fact should be mentioned. Fallopian tube cancer could be discussed together with ovarian cancer.

Chapters concerning breast cancer and endometrial cancer have the paragraph at the end, starting from the words: “In summary”, while others don’t have such summary. Would be better if all chapters have similar summary.

I would suggest to prepare a figure depicting the structure of ERα and its functions. It would be easier to follow than long description in the text. I would also recommend to explain better the difference between ERα and ERβ activity.

Please use epidemiological data from GCO/IARC/WHO, whenever applicable.

I would recommend rejection at this point, just to give the Author enough time to amend the manuscript. If the decision is “major revision” there would be only 10 days for improvement of the manuscript, which is insufficient. I would encourage the Author to resubmit the paper.

 Below are some minor points:

Lane 24: Instead of “… suggesting some of the interplay …” I would say: “…suggesting several mechanisms of cancer development.”

Lane 37: Gynecological system, not: “gynecological tract”

Lane 53: “disruptors” (plural)

Lane 57: “hypothesis” – I wonder if it is right to talk about “hypothesis”, when heavy metals are on the WHO/IARC list of carcinogenic compounds?

Lane 58: “we hypothesize” – there is only one author…

Lane 63: not “gynaecological cancers”, but “female cancers”

Lane 67: what do you mean by “menopausal exposure to exogenous estrogens and progestin”? Better say “HRT for menopausal symptoms”

Lane 67: Why “history of benign breast disease”? What about family history of breast cancer? and breast/ovarian cancer? BRCA mutations, other germline mutations predisposing to breast cancer? This is misleading, when you omit major risk factors.

Lane 75: I would suggest a figure showing the structure or ER and it’s mode of action.

Lane 81: “Regions A/B, located at the N-terminal end, contains transactivation function-1” – rather possess than contain

Lane 96: “This differential expression pattern suggests that ERβ may play distinct roles in these tissues …” There is many data on the role of ERβ, please mention it

Line 105: Please mention, in which species?

Lane 117: “arsenite and cadmium had a significant effect on the incidence and latency of mammary tumors” – what kind of effect?

Lane 123: What do you mean by: “regulation of ER alpha”? Regulation of expression? The same question concerns lane 129.

Lane 179: “This molecular profile is characteristic of basal-like or triple-negative breast cancer” – this is wrong. Neither triple-negative, nor basal-like subtype has expression of ER!

Lane 225: “Ovarian cancer is one of the frequently diagnosed gynaecological cancers” This is not true. Ovarian cancer is not a frequent cancer, however, it has high mortality rate.

Lane 233: “Infertility” is not a risk factor for OC. Treatment for infertility (hormonal stimulation) – is. “oral contraceptive” – pay attention that oral contraceptives use has PROTECTIVE effect against OC (due to less number of ovulatory cycles in the lifetime).

Lanes 261-264: could you mention more about the construction of this study? For me the fragment 261-266 is unclear.

Lane 266: “may contribute to metastasis, and increase cancer risk” – I would mention “cancer risk” first, then “metastasis”

Lane 276-278: “cadmium levels significantly varied depending on the type of tumor and the 276 cycle of chemotherapy administered” – this is unclear

Lane 388: ” Gynaecological cancers are characterized by high prevalence, morbidity, and mortal-388 ity rates.” – This sentence should be changed. Not gyne, but female cancers. And they are very different according to all listed factors – prevalence, morbidity and mortality. You can’t put them into one bag.

Lane 389: “ A growing body of evidence suggests that these cancers are primarily diseases of industrialized societies…” – for this purpose we use epidemiological data (do not say “growing body of evidence”). Secondly, this is not entirely true – e.g. cervical cancer incidence is quite opposite – lower in industrialized countries, higher in undeveloped countries (due to screening, HPV vaccines, etc.)

Reviewer 4 Report

Comments and Suggestions for Authors

Dear Author,

Congratulation for your work and initiative to study and publish this review regarding the contribution of the heavy metals in the etiology of gynaecological cancer. The review is comprehensive and summarize the publish date related to this subject. We still have to do more work in the future regarding the harmful role of heavy metals on our body.

I have just one suggestion to improve your paper: on the page 7 at the section " endometrial cancer" you should put more references.

Kind regards,

Your reviewer

Round 2

Reviewer 3 Report

Comments and Suggestions for Authors

The manuscript has been significantly improved, however, I would still reccomend some changes.

Lines 17-21: I would change sequence of the sentences: first mention the well-known toxic effects of heavy metals, then mention the hormone disrupting activity and underline that this is particularly important in female cancers.

Lane 23: I would repeat here the paragraph from Conclusions (lines 543-545): Despite significant advances, many gaps remain in our understanding of the molecular mechanisms by which heavy metals contribute to cancer development. Addressing these gaps could facilitate the development of more effective strategies for the prevention and treatment of female cancers.

Lane 35, 58 and 60 - you mention here only gynecological system or reproductive system, while you should also mention about breast.

Lane 71 - I would move here (and insert after heading: "Heavy metals and female cancers", while before heading "Breast cancer) the paragraph describing epidemiology of female cancers.

At present it is included in "Conclusions" (lanes 525-533), what is inappropriate, because it is discussed for first time there. You should also add some information about breast cancer and somehow move from this information to heavy metals. My suggestion (can be changed):

"Female cancers, including breast, ovarian, endometrial, and cervical cancers, are associated 525 with substantial morbidity and mortality worldwide, although their prevalence, morbid-526 ity, and mortality vary considerably among different cancer types. Epidemiological data 527 from GLOBOCAN 2022 demonstrate that the incidence and mortality patterns of female 528 cancers vary substantially across different regions and cancer types. Ovarian and endo-529 metrial cancers show higher incidence rates in industrialized countries, which may be at-530 tributed to lifestyle and reproductive factors prevalent in these societies. In contrast, cer-531 vical cancer exhibits a markedly higher incidence and mortality rate in low- and middle-532 income countries, primarily due to inadequate access to effective screening programs and 533 HPV vaccination initiatives. Breast cancer is second most common cancer in woman.  Although the results of breast cancer treatment are improving, the disease negatively affects the quality of life of hundreds of thousands of women around the world. A growing body of evidence suggests that heavy metals and metalloids are associated with the risk of developing various cancers, including female cancers."

Lane 126: "tissue such as the ovary, prostate and endometrium" - only endometrium is "tissue", ovary and prostate that are "organs" - please correct this.

Lane 132: "Similar oncogenic roles are observed in..." rather "effects" than "roles"

Lane 195: "Beyond metalloestrogens, metals and metalloids can also" - I would say: "Beyond estrogen-like activity, metals and metalloids can also"

Lane 235: "acquisition of a cancer cell phenotype independent of ERα activation" - did you mean: "acquisition of a cancer cell phenotype different from ER-positive breast cancer cells"

Lane 356: I would say: "In the context of the etiology of ovarian cancer, which in many cases originates from the epithelium of the fallopian tube (cite e.g.: https://www.sciencedirect.com/science/article/pii/S0002944020304491), there appears...

Conclusions (my proposition; can be changed):

Female cancers are associated with substantial morbidity and mortality worldwide.  Heavy metals can accumulate in tissues and organs, progressively altering their biology and promoting carcinogenesis. The primary mechanism by which heavy metals influence development of female cancers  seems to involve their interaction with estrogen receptors and the disruption of related signaling pathways. Additionally, heavy metals can induce oxidative stress and epigenetic alterations, both of which are critical in the process of tumorigenesis. An overview of the complex interaction network between heavy metals and their numerous molecular targets, summarized in Figure 2, illustrates the multifactorial nature of this influence.

Despite significant advances, many gaps remain in our understanding of the molecular mechanisms by which heavy metals contribute to cancer development. Addressing these gaps could facilitate the development of more effective strategies for the prevention and treatment of female cancers. Given the considerable health risks associated with heavy metals, further research into their role in the pathogenesis of female cancers is essential. 
